# Understanding Sparse Feature Updates in Deep Networks using Iterative Linearisation

## Abstract

Larger and deeper networks generalise well despite their increased capacity to overfit. Understanding why this happens is theoretically and practically important. One recent approach looks at the infinitely wide limits of such networks and their corresponding kernels. However, these theoretical tools cannot fully explain finite networks as the empirical kernel changes significantly during gradient-descent-based training in contrast to infinite networks. In this work, we derive an iterative linearised training method as a novel empirical tool to further investigate this distinction, allowing us to control for sparse (i.e. infrequent) feature updates and quantify the frequency of feature learning needed to achieve comparable performance. We justify iterative linearisation as an interpolation between a finite analog of the infinite width regime, which does not learn features, and standard gradient descent training, which does. Informally, we also show that it is analogous to a damped version of the Gauss-Newton algorithm — a second-order method. We show that in a variety of cases, iterative linearised training surprisingly performs on par with standard training, noting in particular how much less frequent feature learning is required to achieve comparable performance. We also show that feature learning is essential for good performance. Since such feature learning inevitably causes changes in the NTK kernel, we provide direct negative evidence for the NTK theory, which states the NTK kernel remains constant during training.

## 1 Introduction

Deep neural networks perform well on a wide variety of tasks despite their over-parameterisation and capacity to memorise random labels (Zhang et al., 2017), often with improved generalisation behaviour as the number of parameters increases (Nakkiran et al., 2020). This goes contrary to classical beliefs around learning theory and overfitting. This means that some implicit regularisation leads to an inductive bias, which encourages the networks to converge to well-generalising solutions. One approach to understanding this has been to examine infinite width limits of neural networks using the *Neural Tangent Kernel* (NTK) (Jacot et al., 2018; Lee et al., 2019). Interestingly, despite being larger, these often generalise worse than standard neural networks, though with extra tricks, they can perform equivalently under certain scenarios (Lee et al., 2020). Similarly, despite their use for analysis due to having closed-form expressions, they don't predict finite network behaviour very closely. For example, due to the lack of feature learning, they cannot be used for transfer learning, and the empirical NTK (outer product of Jacobians) changes significantly throughout training. In contrast, NTK theory states in the infinite limit that this is constant. This raises important questions about how feature learning (due to the changing kernel) impacts the generalisation behaviour of learnt networks.

To progress towards answering this important question, we empirically examine the effect of freezing feature learning. We introduce *iterative linearisation* — intuitively, an interpolation between standard training and a finite analog of infinite training. This is done by training the proxy linearised model at step $s$ ( $f_{s,t}^{\text{lin}}(x) = f_{\theta_s}(x) + \nabla_\theta f_{\theta_s}(x)^\top (\theta_t - \theta_s)$) for $K$ iterations before *re-linearising* it at step $s + K$. This keeps the Jacobian constant and thus the empirical NTK ($\nabla_\theta f_{\theta_s}(x) \nabla_\theta f_{\theta_s}(x)^\top$) constant, implying that no feature learning is happening.

Contributions

- We introduce a new training algorithm, *iterative linearisation*, that lets us control feature learning and interpolate between lazy (NTK) and rich (feature learning) regimes of training. This opens the way to better study important questions about feature learning in neural networks.
- We show empirically that, in the cases tested, a few feature updates are sufficient to achieve comparable generalisation performance to SGD.
- We provide an example where a few feature updates results in worse generalisation performance when train loss converges faster than features can be learnt.
- We investigate *iterative linearisation* mathematically with low learning rate and large period and show intrinsic connections to second order methods. We provide intuition and empirical evidence as to how damping in second order methods has similar properties to more regular feature updates.

This method allows us to directly quantify the amount of feature learning and its impact. By varying how frequently we re-linearise the model, we show that some feature learning is essential to achieve comparable generalisation to standard stochastic gradient descent, but a small amount of infrequent feature learning is sufficient. We show that this is a valid learning algorithm by drawing connections to the Gauss-Newton algorithm. Furthermore, we show intrinsic links between iterative linearisation and second-order methods, providing a feature learning inspired hypothesis about the importance of damping in such methods.

## 1.1 RELATED WORK

Li et al. (2019) create an enhanced NTK for CIFAR10 with significantly better empirical performance than the standard one, however, it still performs less well than the best neural networks. Yang and Hu (2021) use a different limit to allow feature learning which performs better than finite networks (though it can only be computed exactly in very restricted settings). However, neither of these gives much insight as to how to understand better neural networks using the standard parameterisation. Feature learning has also been studied in the mean field limit Mei et al. (2018); Chizat and Bach (2018) on single hidden layer neural networks where unlike the NTK limit, feature learning still occurs. There are a number of recent works looking at feature learning in this limited setting in a more theoretical way Abbe et al. (2023); Bietti et al. (2023).

Lee et al. (2020) run an empirical study comparing finite and infinite networks under many scenarios and Fort et al. (2020) look at how far SGD training is from fixed-NTK training, and at what point they tend to converge. Lewkowycz et al. (2020) investigate at what points in training the kernel regime applies as a good model of finite network behaviour. Both find better agreement later in training. In contrast, Vyas et al. (2022) look at the continuing changes of the NTK later in training finding that there is continued benefit in some cases.

Chizat et al. (2019) consider a different way to make finite networks closer to their infinite width analogs by scaling in a particular way, finding that as they get closer to their infinite width analogs, they perform less well empirically as they approach this limit.

Much of the second-order optimisation literature considers damping in detail, however it is normally from the perspective of improving numerical stability (Dauphin et al., 2014; Martens, 2010) or optimisation speed (Martens and Grosse, 2015), not understanding feature learning in neural networks.

## 2 PROBLEM FORMULATION

Consider a neural network $f_\theta(x)$ parameterised by weights $\theta$ and a mean squared error loss function[1] $\mathcal{L}(\hat{Y}) = \frac{1}{2}||\hat{Y} - Y||^2$, where we minimise $\mathcal{L}(f_\theta(X))$ for data $X$ and labels $Y$. We can write the change in the function over time under gradient flow with learning rate $\eta$ as:

---

[1] We use MSE for simplicity and compatibility with NTK results here. While this is needed for some NTK results, it does not effect the algorithms we propose where any differentiable loss function can be used — see Appendix A

Figure 1: Spectrum of changing K when always training to convergence. For small values of $K$, it is close enough to (stochastic) gradient descent that it achieves comparable performance, and for very large $K$, it will train until convergence for each linearisation giving the same results for all larger $K$.

$$\dot{\theta}_t = -\eta \nabla f_{\theta_t}(X)^\top (f_{\theta_t}(X) - Y) \tag{1}$$

$$\dot{f}_{\theta_t}(X) = -\eta \hat{\Theta}_t(X, X)(f_{\theta_t}(X) - Y) \quad \text{where} \quad \left[\hat{\Theta}_t\right]_{ij} = \left\langle \frac{\partial f_{\theta_t}(X_i)}{\partial \theta}, \frac{\partial f_{\theta_t}(X_j)}{\partial \theta} \right\rangle \tag{2}$$

It has been shown (Jacot et al., 2018; Lee et al., 2019; Arora et al., 2019) that in the infinite width limit the *empirical neural tangent kernel*, $\hat{\Theta}_t$, converges to a deterministic NTK, $\Theta$. This is a matrix dependent only on architecture and does not change during training. From this perspective, training the infinite width model under gradient flow (or gradient descent with a small step size) is equivalent to training the weight-space linearisation of the neural network (Lee et al., 2019). This raises a number of interesting observations about why this doesn't work well in finite networks and what is different in them. One common hypothesis is that this is due to the lack of enough random features at initialisation, whereas running gradient descent on the full network allows features to be learnt thus reducing the reliance on having enough initial random features. Note that results on pruning at initialisation (Ramanujan et al., 2020) in place of training are distinct from fixing the features like this as hidden layer representations change and hence features are effected by pruning.

## 3 ITERATIVE LINEARISATION

NTK theory says that if the width is large enough, training the weight-space linearisation is equivalent to training the full network (Lee et al., 2019). However in practise training the fully linearised network performs very poorly for practically sized networks (Lee et al., 2020). In this section we propose *iterative linearisation* in order to interpolate between the training of the standard network and the linearised network.

Consider standard (full batch) gradient descent on a neural network with a squared error loss.

$$\theta_{t+1} = \theta_t - \eta \phi_t(f_{\theta_t}(X) - Y) \quad \text{where} \quad \phi_t = \nabla_\theta f_{\theta_t}(X)^\top$$

Here we can think of this as two separate variables we update each step, the weights $\theta_t$ and the features $\phi_t$. However there is no requirement that we always update both, giving rise to the following generalised algorithm:

$$\theta_{t+1} = \theta_t - \eta \nabla_\theta f_{s,t}^{\text{lin}}(X) \tag{3}$$

$$f_{s,t}^{\text{lin}}(x) = f_{\theta_s}(x) + \nabla_\theta f_{\theta_s}(x)^\top (\theta_t - \theta_s) \tag{4}$$

where $s = K * \lfloor \frac{t}{K} \rfloor$ such that every $K$ steps, the neural network $f(\cdot)$ is *re-linearised* through its first order Taylor expansion at the weights $\theta_s$. We note that except at these re-linearisation steps, the features being used do not change.

Using this framework, when $K = 1$ this is simply gradient descent and when $K = \infty$ it is fully linearised training. Other values of $K$ interpolate between these two extremes. See Algorithm 1 for more details. Note that we can also generalise this to not be periodic in terms of when we update $\phi$ so we call this *fixed period* iterative linearisation.

### 3.1 JUSTIFYING ITERATIVE LINEARISATION AS A GENERALISATION OF GAUSS-NEWTON

At first glance, substituting parameters learnt in the proxy linear model back into the full neural network can be surprising and counter-intuitive as a valid learning algorithm. In this section, we

**Input:** learning rate $\eta$, update periodicity $K$, pre-initialised parameters $\theta_0$
**for** *t = 1..steps* **do**
$\quad$ $\theta_t \leftarrow \theta_{t-1} - \eta \nabla \mathcal{L}(f^{\mathrm{lin}}(X; \theta))$;
$\quad$ **if** *t mod K = 0* **then**
$\quad\quad$ $f^{\mathrm{lin}}(X; \theta) \leftarrow f_{\theta_t}(X) + \nabla f_{\theta_t}(X)^\top (\theta_t - \theta)$;
$\quad$ **end**
**end**

**Algorithm 1:** Iterative Linearisation (fixed period)

justify why this is valid; we focus on large values of $K$ as for smaller values, it is very similar to $K = 1$ in terms of learning trajectory.

Consider a large $K$ with a small enough learning rate that gradient descent will converge and give the same solution to gradient flow. In this instance, we fully solve a linear model with gradient flow. This has a closed-form solution, and the following can replace each set of $K$ steps (see Appendix B for a full derivation).

$$\theta_{t+1} = \theta_t - (\phi_t^\top \phi_t)^{-1} \phi^\top (f_{\theta_t}(X) - Y) \tag{5}$$

This is exactly a second-order step using the Generalised Gauss-Newton matrix. Hence, this variant of iterative linearisation is exactly equivalent to the Gauss-Newton algorithm. This exact equivalence relies on a squared error loss, but we can easily generalise to the idea of exactly solving the convex problem that results from any convex loss function on the linearised neural network, though there will no longer be a closed-form solution. From this perspective, we can see the Gauss-Newton algorithm as the special case where we exactly solve iterative linearisation on a squared error loss. As the Gauss-Newton step is a descent direction, we can expect iterative linearisation to be a valid optimisation algorithm that converges to a local optimum.

### 3.2 An approximate measure of feature learning

Consider Algorithm 1, if $K = \infty$ then training only involves Equation (3), this is purely linearised training using random features defined by the neural network's initialisation function. At this step, no feature learning takes place. This is similar to linear models with fixed features – albeit not necessarily random features. From this interpretation, the Jacobian $\phi_t$ are the features we use at time $t$, and $K$ determines how frequently we update these features.

We want to point out that feature learning only happens in Equation (4), but not in Equation (3). This inspires us to call Equation (4) the *feature learning* step and use the frequency of re-linearisation steps as a proxy of feature learning. Furthermore, we conjecture the amount of feature learning from training the proxy model (linearised NN) for $K$ steps to be less than that from training the true model (unlinearised NN) for $K$ steps. Intuitively, this will be true due to the proxy model losing the learning signal information of the neural network. Therefore, after training the linearised model for a few steps, any changes to the features are due to randomness. Figure 6 provides empirical evidence on this view.

**Definition 1 (A proxy for feature learning)** *We define our feature learning proxy as* the number of feature learning updates, *or equivalently the number of times we perform re-linearisations. In $N$ epochs of training using Algorithm (1), this will be $\frac{N}{K}$.*

A useful property of this proxy of of feature learning is that, it is monotonic in the (unobserved) true amount of feature learning (informally, amount of information learnt from data). This is a consequence of iterative linearisation being a generalisation of the Gauss-Newton optimisation algorithm described in Section 3.1. However, this proxy is not linear with respect to to the true amount of feature learning in general.

### 3.3 CONNECTING FEATURE LEARNING AND SECOND ORDER OPTIMISATION USING ITERATIVE LINEARISATION

As shown above, iterative linearisation approaches the Gauss-Newton algorithm as $K$ increases if the learning rate is small enough. We now look at iterative linearisation from the perspective of second order methods.

When using second order methods in practise, *damping* is used. This is a constant multiple of the identity added to the matrix before inverting. This gives the new update below.

$$\theta_{t+1} = \theta_t - (\phi_t^\top \phi_t + \lambda I)^{-1} \phi^\top (f_{\theta_t}(X) - Y) \tag{6}$$

While the solution to the linearised network with squared error loss $\mathcal{L}(\theta) = \|f_{\theta_0}^{\text{lin}}(\theta; X) - Y\|^2$ is given by Equation 5, the solution to the linearised network with loss $\mathcal{L}(\theta) = \|f_{\theta_0}^{\text{lin}}(\theta; X) - Y\|^2 + \lambda\|\theta - \theta_0\|^2$ is given by Equation 6.

We use this to justify a correspondence between damping and smaller $K$ in iterative linearisation and hence with more feature learning. The two methods converge as $\lambda \to 0$ and $K \to \infty$, becoming the Gauss-Newton algorithm. As $\lambda \to \infty$, damped Gauss-Newton approaches gradient flow, while as $K \to 1$, iterative linearisation approaches gradient descent. However, for a small enough learning rate, these are equivalent. Intuitively, as $\lambda$ increases, the regularisation to not move as far in parameter space increases and the linear model is solved less completely. Similarly, with smaller $K$, there are only a limited number of gradient steps; hence the network parameters cannot move as far, and the linearised proxy model is solved less completely. A similar argument can be applied to damping through an explicit step size instead of a multiple of the identity added to the Gauss-Newton matrix.

This raises an interesting hypothesis that the benefit of second-order methods requiring fewer steps may, at the extreme, result in worse generalisation in neural networks due to the lack of feature updates before convergence and provides interesting insight into why damping may be important beyond the numerical stability arguments normally used to justify it. We provide some evidence of this hypothesis in Section 4.2.

## 4 RESULTS

We run experiments with both a simple CNN and a ResNet18 on CIFAR10 to understand the effect of changing feature learning frequency. We would like to point out that we do not aim for state-of-the-art performance (it only gets to maximum $\sim 80\%$ for CIFAR10) as this is not necessary to prove our claims — the important comparison is to contrast large $K$ to small $K$ for iterative linearisation. We use cross-entropy loss in these experiments as, unlike NTK theory, no part of our derivation relied on MSE. In order to improve numerical stability and ensure that the output is a probability distribution during training, we do not linearise the softmax loss function.

### 4.1 SGD PERFORMANCE CAN BE ACHIEVED WITH INFREQUENT FEATURE UPDATES

We compare large $K$ to $K = 1$ to show that only infrequent feature updates are needed in order to achieve comparable generalisation performance. This is true for full batch runs (Figure 2 left), SGD runs (Figure 2 centre), when adding data augmentation (Figure 3) where the final results differ by under 1% (82.3% for $K = 1$ and 81.4% for $K = 5\cdot10^6$). This result also holds when using ResNet18 as can be seen in Figure 4, though with BatchNorm it is far more sensitive due to linearising the normalisation layers. It is extremely clear across these results that we get enough feature learning in 8-12 feature updates to achieve comparable generalisation to hundreds of thousands of feature updates when $K = 1$.

We reproduce and extend the approach taken in Vyas et al. (2022) in Figure 5 and show that using the empirical kernel with $K = \infty$ scales poorly with data, so a complete lack of feature learning hurts generalisation, however, any finite $K$ chosen scales similarly to $K = 1$ if trained until convergence on the training data. This is because $K$ is *too small* to make a difference in whether the amount of feature learning needed for the task is achieved. Specifically, by the time it converges to 100% train accuracy, it has already learnt enough features.

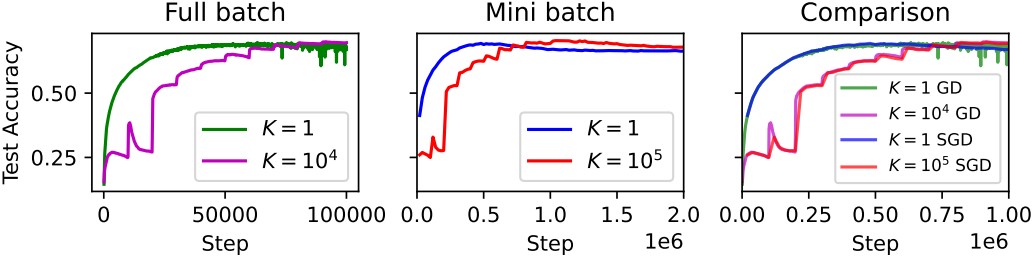

Figure 2: Full batch and mini-batch iterative linearisation for various values of $K$ on a standard CNN architecture on CIFAR10. The left and centre plots compare iterative linearisation to standard training on full-batch and mini-batch gradient descent, respectively. The full batch runs use a learning rate of 1e-3, whereas the mini-batch is scaled down to 1e-4 for stability. As such, we scale $K$ up by a factor of 10, too. The steps of the full batch runs are similarly scaled down by a factor of 10 for the comparison plot on the right.

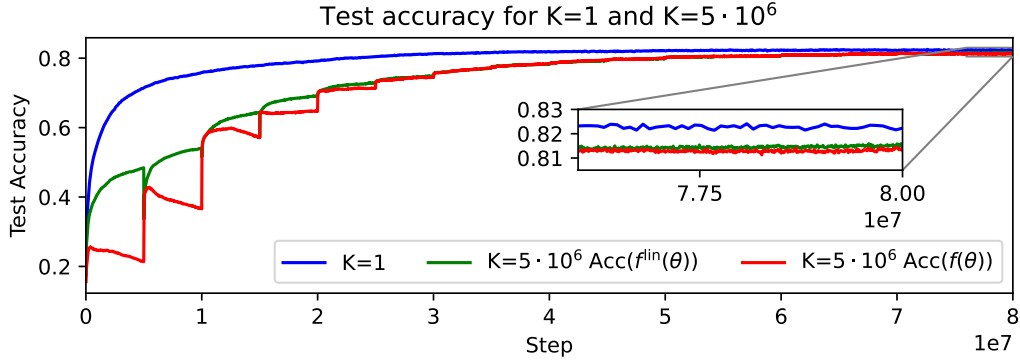

Figure 3: Standard SGD and data augmentation (flips and random crops) for large $K$ on a standard CNN architecture. This shows test accuracy for $K = 1$ as well as both the neural network test accuracy and the test accuracy of the linearised network being trained. As can be seen, the performance is almost equivalent for very large $K$ given enough time to train. Note that as re-linearisation can occur in the middle of an epoch but test performance is evaluated at the end of an epoch, the lines don't drop to the same point, this is simply an artefact of the experimentation, as can be seen on the full-batch graphs.

We also analyse the features which are learnt during this process in two ways. In Figure 6, we use linear probes (Alain and Bengio, 2017) in order to plot the usefulness of the learnt features for the task. This was done through training linear classifiers on the first and second-layer features and plotting test accuracy. We can see that the features for $K = 5 \cdot 10^6$ improve slower than for $K = 1$ and level off at a lower generalisation accuracy. This gives strong evidence that increasing $K$ reduces the amount of feature learning, not simply the frequency. This relationship is not linear, however, and a single step with $K = 1$ results in less feature learning than $10^5$ steps with $K = 10^5$. The performance gap is probably partially due to the less feature learning, but considering the relative gap in linear probe accuracy versus the gap in generalisation, it is clear that less feature learning is not a large hindrance in the case of this combination of model, initialisation and dataset. This measure of features looks at hidden layer activations rather than the Jacobian of the network, so it also provides evidence that both interpretations of *feature learning* align.

In Figure 7, we plot the evolution of the first 5 filters for the first convolutional layer, for both $K = 1$ and $K = 5 \cdot 10^6$, with each column representing 10000 epochs. Each image shows the change in the filter at that epoch and maps from [-0.5,0.5] to [0,1] in order to be plotted. We can clearly see

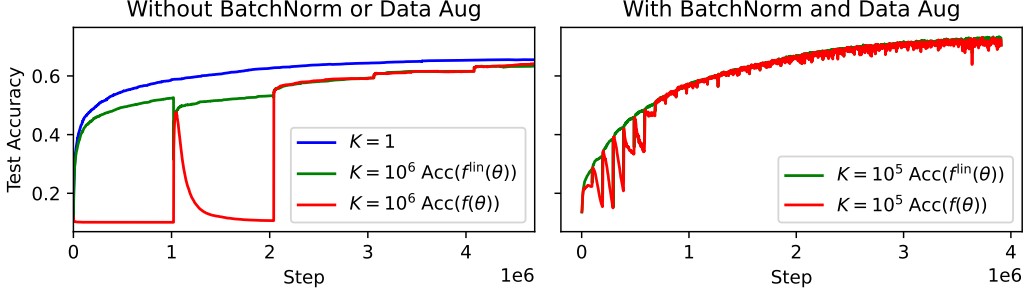

Figure 4: ResNet18 runs with and without BatchNorm and data augmentation. Large $K$ iterative linearisation again achieves similar test performance to SGD. Runs with BatchNorm are far more likely to diverge due to linearising the BatchNorm layer hence why $K$ is much smaller for that run.

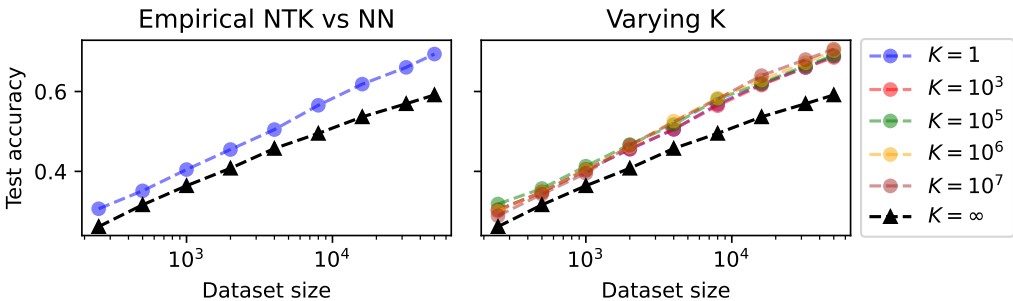

Figure 5: Data scaling behaviour of models as $K$ changes. Here, we show the performance of neural networks with various $K$ when trained on subsets of the training data and trained until convergence. The rate at which performance improves as the dataset size increases is equivalent for all finite $K$ and worse for the case where $K = \infty$ and the features are never updated. Note that the $K = \infty$ runs used a higher learning rate ($\eta$=1e-3 instead of $\eta$=1e-5) for computational reasons, so we include on the left plot a comparison with $K = 1$ with the same higher learning rate to show that this is not relevant to the comparison.

that the filters evolve faster for $K = 1$ and after training is finished for $K = 5 \cdot 10^6$, the features are similar to those of $K = 1$ at about 10% of the way through training.

Overall it seems clear that for this setting, only a few feature updates are needed, however there is still some small generalisation benefit of the more feature learning that occurs with lower $K$.

### 4.2 CONNECTING TO SECOND-ORDER OPTIMISATION METHODS

As covered in Section 3.3, iterative linearisation is closely related to second-order optimisation. We implement the proposed algorithm there where the convex softmax cross-entropy loss of the linearised neural network is solved numerically. To make the graphs more understandable we use a large constant $K$ rather than a varying $K$, intended to make sure that it reaches 100% train accuracy with each re-linearisation and that these are spaced regularly in the graph. We use Adam to solve the linear problem faster than would be possible using vanilla SGD. Here the network converges to a solution which generalises much less well with 61.3% accuracy vs 69.4% for $K = 1$. We ensure that this is not due to using Adam by comparing to an Adam baseline, which achieves over 70%. In this case, the network has achieved a low enough train loss before it has a chance to learn features. This results in less feature learning and less generalisation.

To further investigate the connection between iterative linearisation and damping in the Gauss-Newton algorithm, we show in Figure 9 that the increase in damping — which we have already connected to increased frequency of feature learning — improves generalisation. The bottom part

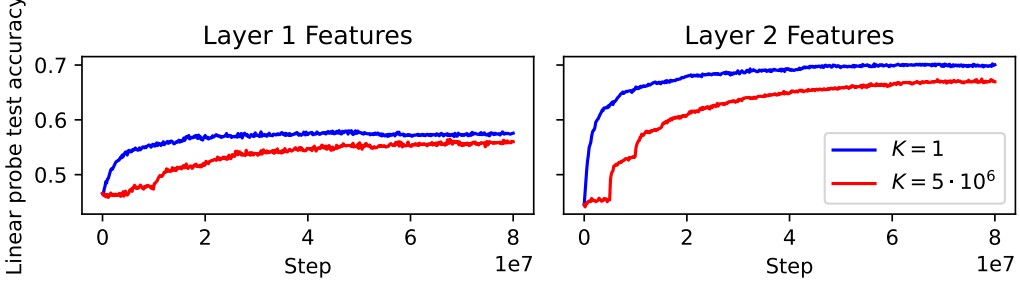

Figure 6: Test accuracy of a linear probe trained on the features of the first (left) and second (right) layers of features. As training progresses, the linear separability of the feature representations learnt at these layers improves, with the large $K$ features improving slower and levelling off at a lower point.

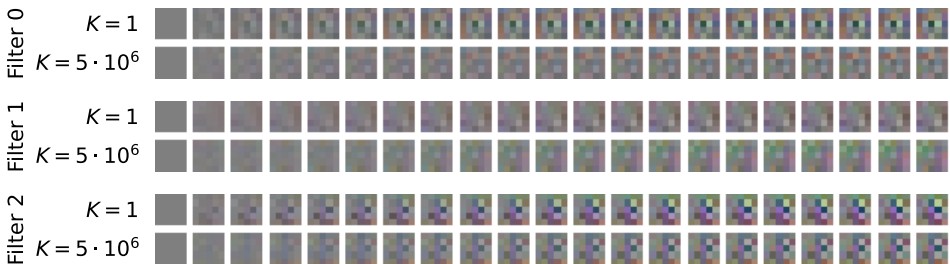

Figure 7: Evolution of the first three convolutional filters of the first layer. Each column represents 10000 epochs and for each filter there is the evolution for $K = 1$ and $K = 5 \cdot 10^6$. Each image maps the [-.5,.5] range of the filter difference from initialisation to [0,1] in order to be plotted. As can be seen, the evolution of the filters for $K = 1$ is much faster and the final filter for $K = 5 \cdot 10^6$ is often similar to $K = 1$ at about 10% of the way through training.

of the figure shows plots of 10 samples trained with each of 4 different damping values and shows visually how the learnt functions become smoother as the damping increases. This gives additional good evidence that too few feature updates during training can result in worse test performance.

## 5 CONCLUSION

This paper has proposed *iterative linearisation*, a novel training algorithm that interpolates between gradient descent on the standard and linearised neural network as a parallel to infinite width vs finite networks. We justify it as a valid learning algorithm with reference to an intrinsic connection to the Gauss-Newton method. We show that by decreasing the frequency of feature updates within iterative linearisation, we can control the amount of feature learning during training, providing a powerful tool to help understand optimisation in neural networks.

In the case of datasets like CIFAR10, we show that a small amount of feature learning is sufficient to achieve comparable test accuracy to SGD across a variety of settings such as full/mini-batch, use of data augmentations, model architecture and dataset size. We also show that *some* feature learning is required for good generalisation, connecting this with the fact that a fixed empirical Neural Tangent Kernel does not learn features and thus does not generalise well. This provides the important insight that while feature learning is necessary, SGD performance can be achieved with significantly less feature learning than expected.

We connect the feature update frequency in iterative linearisation to damping in Gauss-Newton, providing a feature learning-based explanation for damping, backing up this with both theoretical insights connecting them and empirical observations showing the generalisation benefit of increasing damping or decreasing $K$.

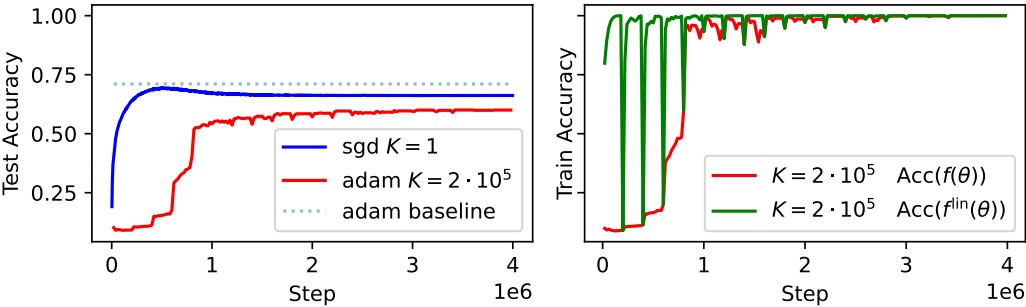

Figure 8: Iterative linearisation using Adam to completely optimise the convex problem each step. Here we compare standard gradient descent ($K = 1$) and a variant of iterative linearisation where the resulting convex objective from each linearisation is fully optimised ($K = 2 \cdot 10^5$) using Adam before re-linearising. In contrast to previous results, this results in worse performance. We note in passing that Adam by itself achieves over 70% on this task so the reduction in performance is due to fully optimising the linearised model and not due to swapping the optimiser.

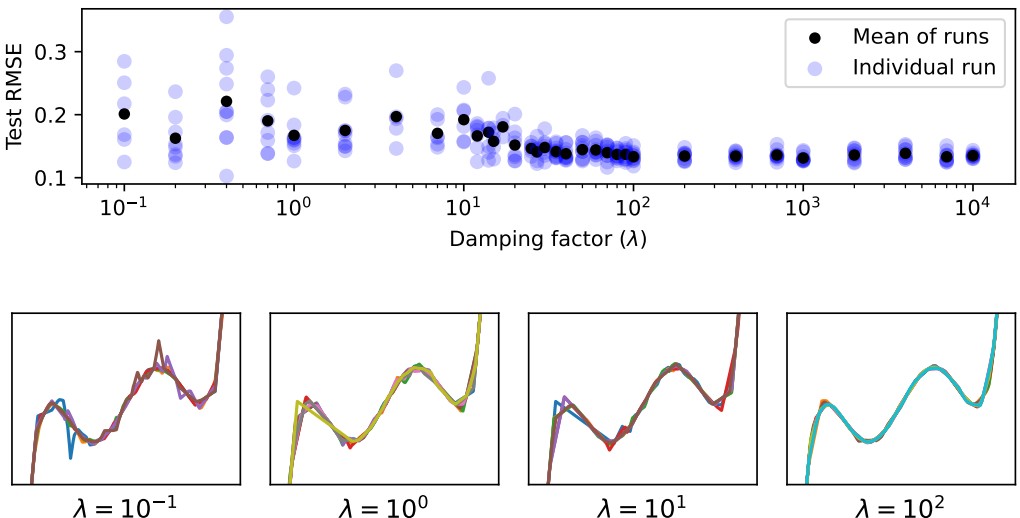

Figure 9: Improvement of generalisation as damping increases. The top plot shows test RMSE as damping is changed (runs with train RMSE over 2.5 are removed), with a reduction in the mean test RMSE and an increase in consistency — in particular note the number of high RMSE points with low damping factor despite the minimums being similar. The bottom plot shows samples of the learnt functions as the damping parameter ($\lambda$) changes, where there is a clear trend towards smoother functions as damping increases.

## 5.1 LIMITATIONS AND FUTURE WORK

Due to computational complexity with the need for small learning rates, most experiments are a single run and on smaller architectures. As such it is possible that these results do not generalise fully to transformers and larger models. For the same reason, we only use CIFAR10. Extending to transformers and more complex datasets would improve the rigour of this line of investigation.

In this paper, we only consider *fixed period* iterative linearisation, where we update the feature vector $\phi$ at regular intervals. However, Fort et al. (2020) showed that the empirical NTK changes faster earlier in training, so it makes sense for $K$ to be more adaptive if this was to be used to inspire more efficient training algorithms. In particular, when fine-tuning large models such as LLMs, there may be a way to improve efficiency by not updating features in this way.

ACKNOWLEDGMENTS

AG and HG would like to thank Javier Antorán, Ross M. Clark, Jihao Andreas Lin, Elre T. Oldewage and Ross Viljoen for their insights and discussions regarding this work. HG acknowledges generous support from Huawei R&D UK.

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

## A    ITERATIVE LINEARISATION WITH A GENERAL LOSS FUNCTION

In Section 3 we show how to get to iterative linearisation from standard gradient under mean squared error loss. The use of mean squared error is more instructive due to its similarities with NTK results, however it is not strictly necessary. For completeness we include here the same idea but for a general loss function $\mathcal{L}(\cdot)$.

Standard gradient descent on a function $f_\theta(\cdot)$ parameterised by $\theta$, with step size $\eta$ and data $X$ can be written as

$$\theta_{t+1} = \theta_t - \eta \nabla_\theta \mathcal{L}(f_{\theta_t}(X))$$

We can apply the chain rule, resulting in

$$\theta_{t+1} = \theta_t - \eta \nabla_\theta f_{\theta_t}(X)\mathcal{L}'(f_{\theta_t}(X))$$

Where $\mathcal{L}'(\cdot)$ is the derivative of $\mathcal{L}(\cdot)$ (in the case of mean squared error, this is the residual: $\mathcal{L}'(\hat{Y}) = \hat{Y} - Y$). Now again using $\phi_t = \nabla_\theta f_{\theta_t}(X)$, we can write this as

$$\theta_{t+1} = \theta_t - \eta \phi_t \mathcal{L}'(f_{\theta_t}(X))$$

With a similar argument to Section 3, we note that we don't need to update the features $\phi_t$ every step, resulting in the following formulation.

$$\theta_{t+1} = \theta_t - \eta \phi_s^{\text{lin}} \mathcal{L}' \left( f_{s,t}^{\text{lin}}(X) - Y \right) \tag{7}$$

$$\phi_s^{\text{lin}} = \nabla_\theta f_{\theta_s}(X) \tag{8}$$

where $s = K * \lfloor \frac{t}{K} \rfloor$

This now lets us use softmax followed by cross-entropy in the loss $\mathcal{L}(\cdot)$ while maintaining the same interpretation, as we do for the MNIST and CIFAR10 results.

## B    CLOSED FORM SOLUTION TO GRADIENT FLOW

Here we show for completeness the derivation of the closed form solution to gradient flow which is used in the equivalence to the Gauss-Newton algorithm. The evolution of the function only depends on the residual so the differential equation can be solved in closed form.

$$\dot{\theta}_t = -\eta \phi^\top (f_{\theta_t}(X) - Y) \tag{9}$$

$$\dot{f_{\theta_t}}(X) = -\eta (\phi\phi^\top)(f_{\theta_t}(X) - Y) \tag{10}$$

$$z_t = f_{\theta_t}(X) - Y \tag{11}$$

$$\dot{z}_t = -\eta(\phi\phi^\top)z_t \tag{12}$$

$$z_t = e^{-\eta(\phi\phi^\top)t}v_s \tag{13}$$

$$f_{\theta_t}(X) = Y + e^{-\eta(\phi\phi^\top)t}(f_{\theta_s}(X) - Y) \tag{14}$$

And for the weights

$$\dot{\theta}_t = -\eta \phi^\top (f_{\theta_t}(X) - Y) \tag{15}$$

$$= -\eta \phi^\top e^{-\eta(\phi\phi^\top)t}(f_{\theta_s}(X) - Y) \tag{16}$$

$$\theta_t = \phi^\top (\phi\phi^\top)^{-1} e^{-\eta(\phi\phi^\top)t}(f_{\theta_s}(X) - Y) + C \tag{17}$$

$$\theta_t - \theta_0 = -\phi^\top (\phi\phi^\top)^{-1} \left(I - e^{-\eta(\phi\phi^\top)t}\right)(f_{\theta_s}(X) - Y) \tag{18}$$

$$\theta_\infty = \theta_0 - \phi^\top (\phi\phi^\top)^{-1}(f_{\theta_s}(X) - Y) \tag{19}$$

This solves it in a kernelised regime, the typical Gauss-Newton step is not in such a regime. If we assume that $(\phi^\top \phi)$ is invertible then we can write $\phi^\top (\phi\phi^\top)^{-1} = (\phi^\top \phi)^{-1}(\phi^\top \phi)\phi^\top (\phi\phi^\top)^{-1} = (\phi^\top \phi)^{-1}\phi^\top$ to get the original formulation.

## C  FURTHER EXPERIMENTAL DETAILS

All CIFAR10 experiments (except those with ResNets in which used a ResNet-18) use a modified variant of LeNet with two convolutional layers, each with kernel size 5 and 50 channels and are followed by dense layers of sizes 120 and 84. All inner activations are ReLU and the output layer uses softmax. This was chosen as simply a larger and more modern version of LeNet. Learning rates are given in plot captions and batch sizes were either 256 or full batch. For Figure 5, optimal early stopping was used.

For the ResNet results in Figure 4 shows performance on a ResNet-18 without BatchNorm or data augmentation (left) with a learning rate of 7e-7 and $K = 10^6$ and on a ResNet-18 with BatchNorm and data augmentation (right) with a learning rate of 1e-5 and $K = 10^5$. While these clearly work similarly, they are computationally expensive and the one with BatchNorm would require a lot of tuning of $K$ and $\eta$ and many days of training to get as clean graphs as we have for a simple CNN.

All 1 dimensional regression tasks use are trained on a quintic function ($\frac{1}{32}x^5 - \frac{1}{2}x^3 + 2x - 1$) — chosen for having nontrivial features and bounded outputs in the [-4, 4] range — with 20 uniformly spaced datapoints used for training data and 1000 uniformly spaced datapoints for the test data. The neural network was a 5 layer MLP with 50 neurons per layer with ReLU activations and squared error loss. They were trained full batch through the Gauss-Newton algorithm described using the given $\lambda$ values for damping.

Experiments were run on a combination of RTX 2080 TI and RTX 3090 graphics cards and took approximately 1500 GPU-hours altogether.

