# OpenReview forum: "Understanding Sparse Feature Updates in Deep Networks using Iterative Linearisation"
_ICLR.cc/2024/Conference — Submitted to ICLR 2024_

### Official Review · Reviewer_XVAB · 2023-10-31

**Soundness:** 2 fair
**Presentation:** 2 fair
**Contribution:** 1 poor
**Rating:** 3
**Confidence:** 4

**Summary:**

The paper proposes *iterative linearization* procedure of neural network training in order to get insights into feature learning process in those networks. The procedure consists of training rounds: having model parameters $\theta_s$ at the beginning of the round, the model is linearized and trained for $K$ iterations (having the gradients $\nabla f_{\theta_s}(x)$ during this time), and then the resulting parameter values $\theta_{s+K}$ are used as linearization point for the next round. Thus, assuming fixed total number of iterations $T$, the procedure interpolates between standard training of a full non-linear model $f_\theta(x)$ at $K=1$ (e.g. features are updated on each iteration) and the training of the fully linearized model $f_{\theta_0}^{lin}(\theta; x)=f_{\theta_0}(x)+\nabla f_{\theta_0}(x)^\top (\theta-\theta_0)$ at $K\geq T$. Also, The authors point to the analogy of iterative linearization with Gauss-Newton second-order optimization with parameter with damping parameter $\lambda$.

Then, the authors empirically examine the performance of the iterative linearization on CIFAR10, with two values of $K$ considered in most of the experiments: $K=1$ and a large enough value $K\sim 10^4 - 10^6$ corresponding to the several $\sim 10$ future updates through the training. From these experiments, the authors conclude that making a few feature updates is sufficient to regain most of the performance of the full feature learning.

**Strengths:**

The main advantage of the paper is the proposed *iterative linearization scheme*. If accurately analyzed, it could exhibit quite an interesting 2D phase diagram of how performance depends on total training time $T$ and feature update frequency $K$.

**Weaknesses:**

Overall, the paper performs a very limited analysis of *iterative linearization*. The only contribution that is convincingly supported is the claim that a few $8-12$ updates are sufficient to the performance comparable to the full training with $K=1$. However, in its current form, this statement does not go much beyond a simple expectation of how interpolation between $K=1$ and $K=\infty$ should look like. An example of a more insightful result would be the whole curve of dependence of a given performance metric on $K$ in the region of its most significant change (probably $K\lesssim 100$ given the provided results). While formally equivalent, the number of feature updates $T/K$ seems to be a more convenient variable than frequency $K$.

The writing of the paper has a feeling of a "flow of thoughts" with lots of completely unsupported statements. Moreover, the paper doesn't have the *contributions*, making it difficult to distill the main claims and how they are supported.

- The discussion of feature learning via pruning in the paragraph after eq. (2) is not clear in its current form.
- The description of iterative linearization in sec. 3 is confusing as it is different from the one described in the introduction (and which was quite clear). In particular, algorithm 1 does not fully describe the iterative linearization procedure, looking as being in the middle of its writing. Eq. (4) is referenced in sec. 3.2 as *feature learning step*, but it only tells that the features of linearized network are gradients.
-  *a proxy measure of feature learning* defined in sec. 3.2. is simply a hyperparameter of the proposed iterative linearization scheme, and therefore is not able to measure in any sense the actual feature learning happening during optimization. Also, the discussion in the paragraph after Def.1 is not clear - e.g. what exactly do you mean by *absolute amount of feature learning*
- It is not clear which linear model is considered in sec. 4.3
- Sec. 4.2 seems absolutely redundant as its only conclusion of poor generalization of linearized models is well known in the literature and already discussed by the authors in the introduction.

(a small typo not affecting the quality of the paper) In figure 3, the neural network and its linearization seem to be swapped in the legend.

**Questions:**

N/A

---

> ### Author Response · Authors · 2023-11-15
>
> Thank you for your thorough and candid review. I think you have found a number of simple phrasing changes that will substantially improve readability so we greatly appreciate that.
>
> To start with, the idea of a phase diagram here is very interesting and not something we looked into in detail, we are running experiments now to investigate this. However, most likely based on what we have seen, it will show that the improvement “per feature update” increases as K increases but the final result remains very similar (the training graph gets closer to going straight up then across). While interesting and a useful graph to include, I don't know that it sheds any new light onto general behaviour.
>
> While we do list the contributions in the introduction, they were in paragraph form. We have split them out into dot points in the updated version and hopefully clarified it. We list these below now to also push back on the claim that our only contribution is the small number of feature updates necessary (which we already believe is a significant contribution for values of K=10^6 as that is not “a simple expectation of the interpolation”).
> * The iterative linearisation algorithm itself is new (to our knowledge), and is definitely new from the context of understanding generalisation and feature learning in neural networks. This opens up an important tool to understand feature learning through better experimentation.
> * We show mathematically that iterative linearisation is equivalent to the Gauss-Newton algorithm with a very large K and very small learning rate limit (when each step is equivalent to running gradient flow to convergence).
> * Generalisation impact of damping in small neural networks: we show empirically that by adjusting damping in Gauss-Newton over multiple orders of magnitude, there is a point where we get a large improvement in generalisation. This is already of independent interest to training neural networks, however combined with our proposed connection to iterative linearisation it provides an important hypothesis connecting two independent areas - namely feature learning and damping in second order optimisation.
> * We show an example with a similar number of feature updates where it does not perform as well due to the training loss getting to zero before there is a chance to learn features.
>
> Regarding the issues mentioned:
> * Feature learning vs pruning: We agree that this is a bit of limited discussion. It was added due to many questions about why this is distinct from ideas about the lottery ticket hypothesis. We have rephrased to remove any claim of the amount of possible options in place of the simpler observation that hidden representations must change under pruning. While this doesn’t give as much intuition, we hope this removes your issue with an unsupported claim.
> Iterative linearisation description: The description in section 3 is equivalent to the one in the introduction but was phrased to make the updating of the features as explicit as possible as that was the common point which was hard to get across. We have rephrased it to be more consistent with the usage in the introduction in the hope that that will help here. Similarly, algorithm 1 was exactly equivalent but we have made the dependence on theta explicit in the closure in the hope that that will improve clarity. Regarding the feature learning step, please notice the subscript s in equation (4) - this is a step to update the features of the network. With the rephrasing we used above, we have now called attention to this in the text as it is no longer an equation to update the features directly. This is a key point and we want to get this idea across as clearly as possible so any more information about which parts you found difficult are helpful.
> * Proxy measure: Perhaps the term 'measure' is inappropriate here and has caused some confusion - we have removed the reference to measure there as it seems to be off putting. Yes this is a hyperparameter, but as we explain, it can control the frequency (and amount - see Figure 6 & 7) of feature learning so is a useful proxy as to the amount of feature learning which occurred. Similarly the term “absolute amount of feature learning” has been replaced by “true amount of feature learning” - the latent true value for which we use our proxy,
> * Linear model in 4.3: the only linear model we consider in this paper is the linearised version of the neural network being trained, however we have made this explicit here to improve clarity
> * Section 4.2 redundancy: we have removed this and covered the details in 4.1
> * Typo in Figure 3: Good catch, we’ve fixed that now :)

---

### Official Review · Reviewer_XaXy · 2023-11-03

**Soundness:** 2 fair
**Presentation:** 2 fair
**Contribution:** 2 fair
**Rating:** 5
**Confidence:** 4

**Summary:**

This paper studies the training of deep networks using iterative linearization. In particular, the algorithm developed in this paper combines the neural network training and NTK linearization methods. The authors show that the feature learning induced by gradient steps is important, and demonstrate that the iterative linearized training can achieve comparable than standard training with fewer steps for feature learning.

**Strengths:**

* This paper proposes an iterative linearization training method that combines the NTK and neural network training.
* This paper performs numerical experiments to demonstrate the importance of feature learning.
* This paper performs experiments to show that feature learning can be made to be less frequent while a comparable test accuracy can still be maintained.

**Weaknesses:**

Overall, this paper provides certain interesting results regarding the effect of neural network training on generalization. However, the main weakness of this paper is that most of the claims are made by numerical experiments without rigorous theoretical justification.

* The authors assume that the gradient descent step is to perform feature learning but do not give detailed justifications. In fact, gradient descent may also memorize the noise to fit training data points. In some special settings (e.g., very wide neural network, specifically designed initialization), gradient descent can also behave similarly to only learning random features.

* Since it is difficult to exactly quantify how much feature learning is performed in gradient descent, there might be some potential concerns by only comparing the number of gradient steps. It is possible that feature learning only happens in a small number of early steps of neural network training, then the "effective number" of feature learning steps could be much smaller than the total iteration numbers.

* As claimed by the authors, iterative linearization is similar to the Gauss-Newton methods, it would be good to also present the results for Gauss-newton method in the experiments.

* In deep neural network training, people typically use a larger learning rate for SGD/GD at least in the early stages, the authors may consider trying a larger learning rate in the experiments, rather than using 1e-3 for all experiments.

**Questions:**

* What happens if applying Gauss-Newton to train the neural network?

* What if using larger learning rates for gradient descent?

* Is there any theoretical explanation to back up the observations?

---

> ### Author Response · Authors · 2023-11-15
>
> Thank you for the review. We believe that there are a few misunderstandings which we call out below, any input in how we can make this clearer in the paper would be greatly appreciated.
>
> * GD step performing feature learning: we completely agree that gradient descent can do memorisation, we only make a claim about when feature learning is happening - as stated in 3.2 “We want to point out that feature learning only happens in Equation (4), but not in Equation (3)”. During equation (3) we are simply using the features that are already there so we can safely say that no feature learning is happening, so all feature learning must be in equation (4). We acknowledge that memorisation and many other phenomena may also occur in the same step but this does not affect any of the later claims or conclusions as we specifically are looking at limiting feature learning - we are ambivalent as to what else happens.
> * Does all the feature learning happen early? This appears not to be the case (e.g. see Vyas et al (2022) in our references). But perhaps it would be helpful for you for us to include a reproduction of this where we train with SGD for various lengths before one-shot linearising? This is a simple plot to make and we can include that in the final version (we have done similar experiments already but not made a clear plot of them)
> * Gauss-Newton on NNs: we agree that this would be an interesting result, it is unfortunately intractable computationally to run a true second order method on a network with > 450k parameters which is why we ran it on smaller networks instead (Figure 9). We do however run an analogue of this in Figure 8 where we train the linearised model to convergence using Adam (if we were using MSE instead of softmax then this would be roughly the same as the Gauss-Newton method with a slight bias due to using Adam). One experiment that we have considered that maybe would assuage this concern for you would be to include the loss term equivalent to the damping factor for a version of the experiment from Figure 8, if you feel this would satisfactorily cover this case then we can run this.
> * Larger learning rates: Even with very small learning rates, we see the same feature learning and generalisation behaviour we wish to study. We acknowledge that this does not reflect standard practice but there is already a lot still to explain with only small learning rates and keeping these small allows us to run experiments that would diverge with larger learning rates.
>
> Questions:
> * Gauss-Newton: As we covered above, we show in Figure 8 that when we do an analogue of Gauss-Newton for softmax networks and approximate the inversion using Adam, we get that it performs worse as would be expected from very little feature learning and the train loss approaching 0 very quickly. We also note that we do train neural networks using Gauss-Newton directly for Figure 9 where it isn’t computationally intractable.
> * Larger learning rates: The proposed algorithm will diverge if the learning rates are too large, but the effects which we intend to study already occur at small learning rates so we do not believe this invalidates our results.
> * Theoretical justifications: This is an empirical paper, while we could make some theoretical justification, when the assumptions of such a justification will clearly not be met, we believe that this is not necessary in order to show our results. We do believe that future work investigating iterative linearisation from a more theoretical direction could provide useful insights but it is not in the scope of this work, nor necessary for what we show.

---

### Official Review · Reviewer_3pTU · 2023-11-04

**Soundness:** 3 good
**Presentation:** 3 good
**Contribution:** 2 fair
**Rating:** 5
**Confidence:** 4

**Summary:**

This paper proposes ``iterative linearization'' as a midpoint between NTK and usual (S)GD update that enables feature learning. Specifically, the algorithm approximates the neural network by linearization every $K$ steps, and updates the proxy model for the next $K$ steps. It has been empirically shown that the comparable performance to usual (S)GD can be obtained even if we increase $K$, while $K=\infty$ (no feature update) is worse than finite $K$. The authors also explain that the connection to the Gauss-Newton algorithm of the proposed algorithm.

**Strengths:**

### The question is well-motivated.

Understanding the gap between NTK and usual (S)GD update is a very important problem. In the infinite width limit the NTK kernel is fixed during training, while practical (S)GD gradually changes the kernel. This paper decompose the role of (S)GD into the feature learning (= update of the kernel, happens every $K$ steps) and optimization on the fixed feature, and tries to identify how many ``times'' of updates of kernel is required to learn features.

### Equivalent performance to (S)GD can be achieved with remarkably few feature updates

The authors experimentally proved that large $K$ can still achieves comparable test accuracy to $K=1$, meaning that feature learning steps can be less frequent than optimization of the linear model. This result gives insights on how feature learning occurs during gradient-based training.

### Connection to the Gauss-Newton algorithm

Iterative linearization is informally connected to the Gauss-Newton algorithm when $K$ is large. This gives some justification to the proposed algorithm.

**Weaknesses:**

### The idea of less frequent feature learning steps is not new

It has now become usual to consider layer-wise training of a two-layer neural network, where the first layer is trained with one step gradient and a large step size, followed by the linear regression of the second-layer parameters. The examples include [Damian et al. (2022)](https://arxiv.org/abs/2206.15144) and [Ba et al. (2022)](https://proceedings.neurips.cc/paper_files/paper/2022/hash/f7e7fabd73b3df96c54a320862afcb78-Abstract-Conference.html). Therefore, in theory, it is not surprising that less frequent updates of the feature suffices to achieve high test accuracy.

### No convergence guarantee

If an algorithm claim itself as a proxy of (S)GD training, it is necessary to have a global convergence guarantee. This paper does not provides any convergence guarantee. Because of its connection to the Gauss-Newton algorithm, it could be possible to derive local convergence, but not global convergence as NTK and the mean-field analysis do. Note that, in order to theoretically justify their argument, we need to show not only the global convergence, but also that increase in $K$ does not slow down the convergence under certain conditions.

### Validity of the proxy measure (Definition 1)

When $K$ is not large enough, the next linearization step comes before the linearized network is fully optimized. Thus, I am not sure whether this value is a number of different features that must be passed through along the way.

### Discussion on the mean-field neural network

Although the network depth is limited to two-layer, the mean-field neural network is an important paradigm in the feature learning analysis. It attributes the optimization of neural networks to the optimization in the measure space and explain the feature learning dynamics [Chizat & Bach (2018)](https://arxiv.org/abs/1805.09545); [Mei et al. (2018)](https://arxiv.org/abs/1804.06561). Recently, several papers [Abbe et al. (2022)](https://arxiv.org/abs/2202.08658); [Abbe et al. (2023)](https://arxiv.org/abs/2302.11055); [Suzuki et al. (2023)](https://openreview.net/forum?id=tj86aGVNb3&referrer=%5BAuthor%20Console%5D(%2Fgroup%3Fid%3DNeurIPS.cc%2F2023%2FConference%2FAuthors%23your-submissions); [Bietti et al. (2023)](https://arxiv.org/abs/2310.19793)) gave convergence and generalization guarantees on specific types of problems (especially polynomials) using the mean-field neural network.

**Questions:**

- Do you think it is possible to give some convergence guarantees on the proposed algorithm?
This does not necessarily for the general functions.

- How can we explicitly know that the feature is learned by the proposed algorithm? Can you evaluate the parameter alignment?

- Is it possible to precisely evaluate the minimum required number of feature updates for some specific problems?

---

> ### Author Response · Authors · 2023-11-15
>
> Thank you for the detailed and well laid-out review!
>
> Before we get to the questions, a few notes on the mentioned weaknesses
>
> * Novelty of less frequent feature updates: Thank you for the links to papers,we had not come across the first paper and are looking forward to reading it in more detail, but I do not believe that the results imply what we have shown here. We had, however, come across the Ba et al paper, it is limited to a single layer as you say which is not the setting which we look at. Additionally, we note that one of our results is to say that when fully solving the linearised problem we get much worse generalisation implying that we are seeing something beyond what has already been studied as that is significantly more than the one step of feature learning in the Ba et al paper.
> * Convergence of iterative linearisation: While we agree that this would be interesting to show, we question why it is necessary. Convergence proofs can be incredibly helpful, however when the goal is to empirically show a phenomena then we argue that they are unnecessary. For the purposes of our results it suffices that it converged in those cases.
> * Proxy measure of feature learning: We agree that this measure of feature learning is not exact - if the linearised model isn’t fully optimised then this will be different. We treat it as a rough proxy for feature learning later in cases where it makes sense. We know of no way to truly measure feature learning accurately in these cases however this allows us to still see correlations.
> * Relation to mean field theory: We agree that the mean field formulation can give us useful insights on feature learning and thank you for some links to newer papers which we hadn’t read previously. Due to the empirical nature of this paper we didn’t attempt to prove what iterative linearisation would accomplish in a mean field setting, nor are we aware of any works which investigate multi-step or second order methods in a mean-field setting. We have now included a discussion in the related work section
>
> With regards to the questions mentioned:
> * Convergence guarantees: we believe it would be possible, but beyond the scope of this work (see above for details). Specifically the form would likely have to be, for a given K and problem, there exists a learning rate such that convergence is guaranteed.
> * Regarding both the second and third questions, these both assume some kind of ground truth for measuring features and how far we are from some kind of “optimal features”. The second question wasn't completely clear to us, but there is very little way to know anything for sure about these features. This is why iterative linearisation and our proxy measure are useful - they are not exact but we should expect them to be correlated such that we can draw useful conclusions.
> * Regarding the third question, we can possibly do something to investigate the minimum number of features needed to learn, however due to not having any ground truth about how much features are updated or what the “optimal features” are, decisions about the K value to use become very important (e.g. see Figure 8 for why we cannot just use extremely large K) and have no obvious solution

---

> > ### Comment · Reviewer_3pTU · 2023-11-15
> >
> > Thanks for the reply. I agree that iterative this paper demonstrates interesting phenomena regarding feature learning beyond two-layer neural networks, while I disagree on the following points.
> >
> > - "we are seeing something beyond what has already been studied as that is significantly more than the one step of feature learning in the Ba et al paper."
> >
> > 　 I think a long history of discussions on NTK and comparison with kernel methods have already proven this fact.
> >
> > - "we should expect them to be correlated such that we can draw useful conclusions"
> >
> > 　If $K$ is small, then solving a linear system in each feature is not complete. If $K$ is infinity, then each linear system can be exactly solved. I think this difference have a significant affect even if such a correlation exists. I want to know how to reduce or correct such an effect.
> >
> > Therefore, I would like to maintain my score currently.

---

> > > ### Author Response · Authors · 2023-11-17
> > >
> > > Thank you for your response and clarifying the parts that you still disagree with.
> > >
> > > Apologies, I believe I know where the misunderstanding occurred here. Your first point was responding to something in our response which was phrased very ambiguously in hindsight. We were referring to Figure 8 which shows that continuously solving the linearised model to completion results in worse performance. This is distinct from the NTK regime which as you mentioned has been studied extensively. With this algorithm we do learn features (and more than one feature step) yet still result in much worse performance. This shows that either not enough features are learned (e.g. due to train loss reaching 0 before there is a chance to learn them), or somehow the wrong features are learned. This feature learning regime with multiple steps is what we were referring to being beyond what the Ba et al paper studied, and has definitely not been studied in the NTK and kernel literature.
> > >
> > > We completely agree that the amount of feature learning per update depends on K. As far as we are aware, there does not currently exist a way to properly measure features or feature learning, but using this method we can learn a lot regardless. We also want to know how to do such a measurement exactly but it is unfortunately well beyond state-of-the-art to do so and as such we do not attempt an exact measure. Using our method, we can still show that reducing the feature learning does not significantly affect performance here despite it being an inexact measure (see Figures 6 and 7 to see that by most definitions less feature learning is happening). We believe that this is still an important result despite the limitations you mention.

---

> ### Comment · Reviewer_3pTU · 2023-11-17
>
> Thank you for your further clarification.
>
> As for Figure 8, why did you use SGD for $K=1$ and Adam for $K\gg 1$? In order to draw a fair comparison between different $K$, I think using the same algorithm is necessary. I am still not convinced about whether Figure 8 brings us new insights compared to NTK and existing feature learning papers.

---

> > ### Author Response · Authors · 2023-11-17
> >
> > It is unclear which part of the decision you have an issue with so let me clarify the relevant decision reasoning:
> >
> > **Why use Adam at all?**
> > We use Adam for faster convergence in order to better simulate the second order method it would approach with large K (SGD takes too long to converge and inverting the kernel is infeasible computationally due to the O(n^3) cost).
> >
> > **Why compare Adam with SGD**
> > We believe that SGD is a better baseline still as we are trying to compare to how it works with SGD and large K elsewhere and wish not to include other inductive biases that Adam brings with it. Adam already improves generalisation and it is unclear that that inductive bias will carry onto the setting where it is used to fully solve the linearized model.
> >
> > **What if Adam as a baseline would give different results?**
> > You will notice that for completeness on that same graph and in the caption we already *also* include Adam with K=1 as a baseline to be sure this isn't the case.
> >
> > **Why is the Adam inclusion a maximum accuracy rather than a training trajectory?**
> > We include it as a horizontal line rather than the training trajectory as of course Adam converges much faster and this makes it easier to see how much better it is compared to SGD. Additionally it makes it easier to compare the two methods we actually wish to compare.
> >
> > **New insights** If you believe that this was already predicted by other papers perhaps you could point us to this? I don't believe it is implied by the papers you linked to my understanding of them.

---

> > > ### Comment · Reviewer_3pTU · 2023-11-17
> > >
> > > I understood. The baseline corresponds to Adam with $K=1$, right?

---

> > > > ### Author Response · Authors · 2023-11-17
> > > >
> > > > Yes, perhaps that was the bit that was confusing? K=1 is of course equivalent to the standard algorithm. We use the term baseline as we aren't showing the full training trajectory, perhaps "Adam K=1 baseline" is less ambiguous?

---

> > > > > ### Comment · Reviewer_3pTU · 2023-11-17
> > > > >
> > > > > I think so. Or just plotting the trajectory for Adam with $K=1$ would also be OK.
> > > > >
> > > > > I think, understanding the complexity of problems that machine learning is addressing in practice is an important research theme, and the main message that there is no need to pass through thousands of intermediate features during training is intriguing (although already rigorously demonstrated for limited classes of problems in theory as I noted). On the other hand, it's undeniable that there is a lack of quantitative metrics other than "$K$", which makes me hesitate to improve my score. I would like to maintain this score.

---

### Meta-Review · Area_Chair_igYG · 2023-12-05

**Metareview:**

This paper tries to understand feature-learning in neural networks by considering an iterative linearization algorithm. The iterative linearization algorithm interpolates between NTK and standard training, controlled by a parameter K. The paper experimented with K = 1 (standard training) and larger K to argue that feature learning is important, and updating the feature several times is sufficient to get most of the benefit. On the theoretical side, the paper connects the iterative linearization idea to Gauss-Newton methods. While reviewers find the connection to Gauss-Newton method and the idea of iterative linearization interesting, there are some concerns about lack of theoretical guarantees for the iterative linearization algorithms, and concerns on experiments not showing the whole range of K. The authors should revise the paper based on reviews for future submission.

The updated version of the paper had acknowledgement which breaks anonymity. Please be more careful for future conferences.

**Justification For Why Not Higher Score:**

While the idea seems interesting, different reviewers all have some concerns and it seems like the paper would benefit a lot from a major revision.

**Justification For Why Not Lower Score:**

N/A

---

### Decision · Program_Chairs · 2024-01-16

Reject